# Increasing Influenza Vaccination Uptake by Sending Reminders: A Representative Cross-Sectional Study on the Preferences of Italian Adults

**DOI:** 10.3390/vaccines11101601

**Published:** 2023-10-16

**Authors:** Alexander Domnich, Riccardo Grassi, Elettra Fallani, Giulia Costantini, Donatella Panatto, Matilde Ogliastro, Marco Salvatore, Maura Cambiaggi, Alessandro Vasco, Andrea Orsi, Giancarlo Icardi

**Affiliations:** 1Hygiene Unit, San Martino Policlinico Hospital-IRCCS for Oncology and Neurosciences, 16132 Genoa, Italy; andrea.orsi@unige.it (A.O.); icardi@unige.it (G.I.); 2SWG S.p.A., 34133 Trieste, Italy; riccardo.grassi@swg.it (R.G.); giulia.costantini@swg.it (G.C.); 3CSL Seqirus, 53035 Monteriggioni, Italy; elettra.fallani@seqirus.com (E.F.); marco.salvatore@seqirus.it (M.S.); maura.cambiaggi@seqirus.com (M.C.); alessandro.vasco@seqirus.com (A.V.); 4Department of Life Sciences, University of Siena, 53100 Siena, Italy; 5Department of Health Sciences (DISSAL), University of Genoa, 16132 Genoa, Italy; panatto@unige.it (D.P.); matilde.ogliastro@hsanmartino.it (M.O.); 6Interuniversity Research Center on Influenza and Other Transmissible Infections (CIRI-IT), 16132 Genoa, Italy

**Keywords:** influenza, vaccination, influenza vaccine, vaccination uptake, remainder, survey, Italy

## Abstract

Evidence from countries that achieved a high seasonal influenza vaccination (SIV) coverage suggests that reminders to get vaccinated may increase SIV uptake. The goal of this study was to explore the experience and attitudes of Italian adults toward an active invitation to receive SIV, triggered by different sources and delivered via different communication channels, and to assess the projected benefits of this strategy. A cross-sectional survey on a representative sample of Italian adults was conducted by using computer-assisted web interviewing. Responses from 2513 subjects were analyzed. A total of 52.2% of individuals previously received invitations to undergo SIV and compared with people who did not receive any reminder were three times more likely (68.2% vs. 22.2%) to be vaccinated in the last season. Compared with other sources, reminders sent by general practitioners (GPs) were perceived as the most attractive. As for communication channels, most participants preferred text/instant messaging (24.6%) or email (27.2%), suggesting an acceleration in the Italian digital transformation triggered by the COVID-19 pandemic. Conversely, traditional postal letters or phone calls were preferred by only 17.0% and 8.6% of respondents, respectively. Reminders sent by GPs via text/instant messages or email are a valuable option for increasing SIV uptake among Italian adults.

## 1. Introduction

Seasonal influenza vaccination (SIV) is an important public health strategy to prevent severe disease and several population groups, including older adults, subjects with underlying health conditions, pregnant women, children, and healthcare workers, may benefit from annual immunization [1]. Despite these direct benefits for healthcare systems and the overall welfare of society, SIV coverage is still insufficient in most industrialized and developing countries [2].

Strategies to increase SIV uptake may be broadly summarized as interventions that (i) increase community demand, (ii) enhance access, and (iii) target healthcare providers or systems [3]. The central role in improving people’s demand for SIV (also through addressing vaccination hesitancy) may be exemplified by the effect of the ongoing COVID-19 pandemic on SIV coverage. During the first pandemic phases (and before COVID-19 vaccines became available), public acceptance of SIV increased and previously eligible but unvaccinated people received their SIV for the first time [4,5]. Conversely, once mass COVID-19 vaccination campaigns were rolled out, a significant decrease in SIV uptake has been reported [6]. This polarizing effect may be easily traced in Italy: while the 2020/21 SIV coverage in older adults aged ≥65 years registered a relative gain of 20% (passing from 54.6% in the 2019/20 season to 65.3% in the 2020/21 season), in the 2021/22 season, the SIV coverage dropped to 58.1% with an 11% relative decrease [7].

In Europe, some countries like the United Kingdom (UK) and the Netherlands are more successful in approaching the minimum required SIV coverage in at-risk populations of 75%, especially in older adults [8]. These two benchmark countries were the first to implement comprehensive national guidelines on the roll-out of SIV campaigns, which were developed with a strong endorsement of general practitioners (GPs) [9]. The best practices from these two countries suggest that the identification followed by a personal written notification sent to all eligible individuals have the largest effect on increasing SIV uptake [9]. Accordingly, following a reform of the vaccination policy in the Netherlands, all individuals aged ≥65 years (and also those turning 65 between September and May) receive a personalized invitation letter for a free SIV. Moreover, an additional stock of vaccines is provided to GPs to further increase the opportunity to vaccinate all eligible people [10]. A large UK survey of GPs [11] has shown that sending personal invitations for all at-risk patients (not just catch-up invitations to those who did not respond to an initial general publicity campaign) was associated with the highest SIV uptake among older adults. Of note, interventions based on traditional paper-based or electronic invitations to attend for SIV have among the lowest total costs [12].

In Italy, SIV is currently offered free of charge to older adults aged ≥60 years, pregnant women, subjects with underlying health conditions, children aged 6 months to 6 years, workers at high risk of exposure (e.g., healthcare workers), other professionals of primary public importance (e.g., police), and some other categories. However, SIV is also recommended (but not reimbursed) for all other population groups [13]. The SIV campaign usually starts in mid-October and most doses are administered by GPs, who are remunerated for each vaccination performed [14]. On the other hand, there is no nationwide active invitation program, and such initiatives are mobilized only by some regional health departments (HDs) or local health units (LHUs). Indeed, both SIV uptake and associated policies in Italian regions are highly inhomogeneous [14,15].

The available implementation research converges on the idea that targeted and proactive interventions may increase SIV uptake [16]. It is also known that public trust in SIV-related information varies by information source [5] and some people prefer one communication channel over others [17]. In this study, we aimed to explore the projected effectiveness and attractiveness of the actively sent reminders to undergo SIV, triggered by different sources and delivered via different communication channels, in a nationally representative sample of Italian adults. In particular, there were two main study hypotheses. Based on the above-described UK study [11], we first hypothesized that subjects who had previously received any form of reminder to get SIV would show a greater vaccination uptake and that people’s preferences regarding different reminder sources varies. Our second hypothesis was that people’s preferences regarding different communication channels differ [18,19].

## 2. Materials and Methods

### 2.1. Study Design and Procedures

This cross-sectional study was conducted between 24 October and 10 November 2022 and represents the fourth wave of a longitudinal computer-assisted web interviewing (CAWI) survey, which was established in 2020 with the aim to monitor changes in the knowledge, attitudes, and practices (KAP) on influenza and SIV in a panel of Italian adults [5]. The inclusion criteria were as follows: (i) age ≥18 years, (ii) Internet access, (iii) residence in Italy, and (iv) voluntary informed consent. Each survey round aimed to reach at least 2000 responses. For the present survey, a total of 3247 invitations were sent. These were selected from a pool of approximately 60,000 well-characterized individuals registered in the SWG database. In order to be representative of the adult Italian population, the selection was performed in a two-stage probabilistic quota modality, the details of which may be assessed elsewhere [5]. The questionnaires used for each survey wave were composed of both recurring core items on SIV-related KAP and novel items introduced each time in order to reflect changes in influenza epidemiology and preventive strategies. Both recurring questionnaire items and results of the previous survey waves may be assessed in our previous publications [5,20,21]. This study is instead focused on the items introduced for the first time during the fourth survey wave and are described later in the text.

Participants, who were active members of the SWG dataset, were invited to participate via email. This letter contained general information about the study aim and execution modality and a direct link to the password-protected survey. Before starting the survey, all participants were informed that participation in the study was voluntary, the responses provided would be analyzed in anonymized form, and who the data processor and owner was. After that, all participants provided their written informed consent. The survey had no time limits, a clearly visible progress indicator, one item per screen, and all items were mandatory to reply.

This non-interventional, opinion-based web survey was conducted in accordance with all applicable Italian laws and regulations, including the General Data Protection Regulation.

### 2.2. Study Outcomes

The past experience with receiving reminders to get SIV was measured on a single-choice matrix item “Have you ever received an invitation to get a flu shot delivered to you by…” (i) your GP; (ii) other specialist physicians you are in contact with; (iii) your pharmacist; (iv) your LHU; (v) HD of your region; (vi) your relatives or friends. The responses were coded as (1) Yes and (0) No. For this item, the independent binary variable of interest was the past season (2021/22) SIV uptake (1 = vaccinated). To further confirm or reject the first study hypothesis, we also measured the participants’ attitudes toward SIV and associated reminders for the upcoming 2022/23 season. In particular, we asked subjects to reply on a matrix/rating scale item entitled “How would you judge a personal invitation to get a flu shot delivered to you by…” with the same six response options. Each of these response options was ranked on a 5-point Likert scale (5: Strongly positively; 4: Positively; 3: Neither positively nor negatively; 2: Negatively; 1: Strongly negatively). For this item, the predictor of interest was the intention to receive the 2022/23 SIV (Do you intend to have a flu shot in the upcoming season?), which was measured on a 5-point Likert scale (5: Yes definitely; 4: Probably yes; 3: I don’t know; 2: Probably not; 1: Definitely not).

To test the second hypothesis on the different preferences regarding different communication channels, subjects were asked to indicate a preferred channel for this personal invitation, by selecting one of the following: (i) phone call; (ii) postal letter; (iii) email; (iv) text/instant message on mobile phone; (v) I don’t want to receive any invitation.

### 2.3. Study Variables

Sociodemographic characteristics included sex, age, place of residence, and socioeconomic status (SES). Regions of residence were categorized into three macro-areas of North (Aosta Valley, Liguria, Lombardy, Piedmont, Emilia-Romagna, Friuli-Venezia Giulia, Trentino-South Tyrol, and Veneto), Center (Lazio, Marche, Tuscany, and Umbria), and South (Abruzzo, Apulia, Basilicata, Calabria, Campania, Molise, Sicily, and Sardinia). SES was assessed on the dimensions of education background, personal income, and occupation pattern. In particular, three levels of education were distinguished, namely low (middle school or lower), medium (high/secondary or vocational school), and high (university degree or higher). Perceived income was classified into low, lower than average, average, higher than average, high, and no personal income. Finally, people’s occupation pattern could be one of the following: employed, student, housekeeper, retired, unemployed, or other.

### 2.4. Statistical Analysis

Categorical variables were expressed as percentages with Clopper–Pearson exact 95% confidence intervals (CIs), while continuous variables were reported as medians with interquartile ranges (IQRs). Proportions were compared by means of the Chi-square test. Cochran Q with Bonferroni-corrected post-hoc McNemar tests were used to verify the null hypothesis on the equal distribution of Likert scale-based variables. As SIV in Italy is currently recommended for all older adults aged ≥60 years [13], a subgroup analysis by age (18–59 vs. ≥60 years) was also conducted. To correct for potential confounders, multivariable logistic regression was used to obtain adjusted odds ratios (aORs) on the association between SIV uptake and past receipt or potential attractiveness of reminders to get vaccinated. During the model fitting, a strong collinearity (variance inflation factors > 10) was observed between the nominal variables of occupation pattern and perceived income. Considering that the latter explained more variance, we retained the variable of income in all adjusted models. When the Likert scale-based outcome variable of the likelihood of being administered the 2022/23 SIV was modelled in the ordinal logistic regression, a significant (Brant test: *p* < 0.001) violation of the proportional odds assumption was observed. This variable was therefore dichotomized into (0) will unlikely get vaccinated (responses “I don’t know”, “Probably not”, and “Definitely not”) and (1) likely get vaccinated (responses “Probably yes” and “Yes definitely”). The robustness of the base case model was verified in a sensitivity analysis by changing the classification rule, that is the response option “Probably yes” was moved to the category (0).

All statistical analyses were carried out in R software (packages “stats”, “PropCIs”, “MASS”, “car”, “rstatix”, and “brant”) v. 4.2.2 (R Foundation for Statistical Computing, Vienna, Austria).

## 3. Results

### 3.1. Characteristics of the Study Participants

Of the 3247 invitations sent, a total of 2515 unique responses were received (response rate of 77.5%). Non-responders were similar to responders in terms of sex and macro-area of residence but were younger (60.3% of non-responders were 18–34 years). Two (0.1%) subjects were residing abroad and were excluded. In summary, responses from 2513 subjects were analyzed (Figure 1).

The principal sociodemographic characteristics of the study participants are reported in Table 1. Briefly, their median age was 51 (IQR 37–66, range 18–84) years and both sexes were approximately equally distributed. Most subjects resided in Northern Italy, achieved at least middle school, were employed, and declared an average or higher than average income. A total of 46.4% reported receipt of the 2021/22 SIV. As expected, the self-reported 2021/22 SIV uptake was significantly (*p* < 0.001) higher in participants aged ≥60 years (68.2%; 95% CI: 65.0–71.3%) than those aged 18–59 years (34.4%; 95% CI: 32.1–36.7%).

### 3.2. Association between Influenza Vaccination Uptake and Influenza Vaccination Reminder

Approximately half of participants (52.6%; 95% CI: 50.6–54.6%) had previously received at least one reminder to get vaccinated with SIV. Receipt of any reminder was significantly higher in older adults aged ≥60 years (68.8%; 95% CI: 65.6–71.8%) than in younger adults aged 18–59 years (43.7%; 95% CI: 41.3–46.2%). The 2021/22 SIV coverage among individuals who were invited (68.2%; 95% CI: 65.6–70.7%) to get vaccinated was about three times higher than among those who did not receive any reminder (22.2%; 95% CI: 19.8–24.6%) with an aOR of 6.47 (95% CI: 5.35–7.83). As shown in Table 2, most reminders came from participants’ GPs (39.3%; 95% CI: 37.4–41.2%), followed by friends or relatives (22.8%; 95% CI: 21.1–24.5%) and specialist physicians (16.2%; 95% CI: 14.8–17.7%). Reminders from LHUs (13.0%; 95% CI: 11.7–14.4%), HDs (12.5%; 95% CI: 11.2–13.8%), and pharmacists (11.5%; 95% CI: 10.3–12.8%) were less prevalent. However, in the fully adjusted model, only invitations made by GPs, specialist physicians, and LHUs were associated with the past season SIV receipt (Table 2).

Regarding the attractiveness (responses “positively” or “strongly positively”) of single reminder sources, people’s ratings were unequally distributed (*p* < 0.001) in the following descending order: GP, LHU, specialist physician, HD, pharmacist, and relatives/friends (Figure 2). All pairwise comparisons were highly significant (*p* < 0.001) except that between specialist physician and HD (*p* > 0.99).

A total of 46.9% (95% CI: 45.0–48.9%) of respondents declared their willingness (29.1% and 17.8% replied “Yes definitely” or “Probably yes”, respectively) to receive the 2022/23 SIV. Compared with younger adults (70.2%; 95% CI: 67.1–73.2%), this proportion was higher (*p* < 0.001) among ≥60-year-olds (34.1%; 95% CI: 31.8–36.5%). As shown in Table 3 (Model 1), each 1-point Likert scale increase in the perceived attractiveness of reminders from a GP or LHU was associated with an 81% and 51% increase in the odds of likelihood of receiving the 2022/23 SIV. Other reminder sources did not reach *α* < 0.05. The results of the sensitivity analysis, when only people who would definitely receive the 2022/23 SIV were considered as a success (Model 2), were similar to the base case, although a reminder delivered via their own pharmacist turned statistically significant with an aOR of 1.30 (95% CI: 1.02–1.68) (Table 3).

### 3.3. Preferences on the Reminder Delivery Channel

Digital communication channels, such as text/instant messaging (24.6%) and email (27.2%), were preferred by about half of respondents. Traditional postal letters (17.0%) or phone calls (8.6%) were less preferred. By contrast, 22.6% of individuals did not want to receive any reminder. When analyzed by age group, it emerged that compared with younger adults, significantly more (*p* < 0.001) subjects aged ≥60 years preferred text/instant messages (30.0% vs. 21.6%). Conversely, a significantly (*p* = 0.005) higher proportion of younger adults (24.4% vs. 19.4%) did not want to receive any reminder (Table 4). On considering that most individuals preferred digital communication channels, a post-hoc analysis on the comparison between subjects who preferred email and text messages was performed. When adjusted for the previously received communications and past season vaccination, subjects with the highest income (aOR high income vs. low income 4.56, *p* = 0.030) and those living in Central Italy (aOR Central Italy versus Northern Italy 1.54, *p* = 0.006) preferred email over text messaging. Of note, no differences between sexes, age groups, education level, and previous season SIV uptake emerged, suggesting that both email and text messaging would almost equally reach the target populations.

## 4. Discussion

This is the first Italian study to investigate the public experience and perception of reminders to get vaccinated against seasonal influenza and some important correlates of this have been established. The main study strength is a large sample size of a representative and well-characterized cohort of Italian adults. Here, we demonstrated that half of Italian adults have previously been exposed to some form of reminders to get SIV and these individuals showed significantly higher odds of being vaccinated. Analogously, intentions to get the next season SIV were higher among subjects who perceived more attractive reminders sent by their GP, even when adjusted for the previous season vaccination, age, and other confounders. We therefore confirmed our first hypothesis. We also validated our second hypothesis on the differences in people’s preferences regarding various communication channels: digital channels were favored by most participants.

The principal source of this invitation was participants’ GPs who, among other sources, showed the highest effect size on the past SIV receipt. This finding is in line with the results reported by Dexter et al. [11] who documented the highest (*p* = 0.003) SIV coverage among older adults who received a personal invitation from their GPs. Indeed, a systematic review by Kohlhammer et al. [22] highlighted that the recommendation by GPs is among the strongest positive predictors of SIV. Analogously, GPs were attributed a comparably high ranking as a source of future reminders and subjects who assigned higher scores to GPs were more prone to be vaccinated in the next season. Although SIV hesitancy among Italian GPs seems uncommon and most of them implement some initiatives to engage proactively with their patients [23,24], influenza- and SIV-related knowledge among GPs may be suboptimal. For instance, Vezzosi et al. [23] have reported that only 38.9% of GPs in Parma (Northern Italy) were aware of the minimal recommended SIV coverage rate in at-risk groups of 75%. Considering both a steady progress in the development of novel SIV formulations [25] and increasing availability of high-level evidence on the effectiveness and safety of SIV, national/regional public health authorities, scientific societies, and GP associations should ensure effective forms of continuous medical education activities on the topic, in which a maximum number of GPs are incentivized to take part. In summary, our results confirm the central role of GPs in SIV-related decision-making and underline that future health promotion and social marketing interventions to increase SIV coverage rates in Italy should not be planned or executed without the endorsement of GPs.

Our second major finding is that the majority of Italian adults preferred digital channels like mobile phone messages (24.6%) or emails (27.2%), while the proportion of those who preferred more traditional phone calls (8.6%) or postal letters (17.0%) was substantially lower. This may also indicate an acceleration in the communication paradigm shift toward digital technologies; indeed, the COVID-19 pandemic has sped up the digital transformation of the Italian public service [26]. The available systematic evidence [27,28] converges on the idea that eHealth/mHealth reminders to increase vaccination uptake are, overall, effective and cost-effective when compared with “do nothing” strategies. For example, a randomized controlled trial (RCT) on 12,354 at-risk subjects [29] found that compared to the non-intervention group, individuals who received a text message showed a 39% increase in SIV uptake. However, when comparing the effectiveness of single traditional and digital channels, some discrepancies emerge. Thus, the vaccination completion rate among United States (US) adolescents was 32.1% among those reached via text messaging, as compared with 23.0% and 20.8% among those contacted via postal letter or email, respectively. Of note, the average costs were $4.65 per postal letter and $3.09 per either email or text message [30]. On the other hand, a recent large RCT [31] has documented no detectable increase in COVID-19 vaccination uptake among US adults receiving text messaging compared with telephone calls only. These apparent inconsistencies are likely driven by a number of factors, including the study design, healthcare model in which the study was carried out, type of vaccination, and target population. Interestingly, when individuals who preferred to be invited via email or text messaging were compared directly, no differences in terms of their sex, age, educational background, or previous season vaccination emerged. This finding is of a certain importance, especially for the universal healthcare models like in Italy, as it may signify an almost equal reachability of the principal target populations. Providing that both email and text messaging were preferred in similar proportions, we believe that based on the available infrastructure and operational complexities, single Italian regions may opt for one or another channel. Although our study did not allow for establishing whether a simultaneous adoption of both email and text messaging could have an additive effect, the previous UK experience [11] has shown that using two communication channels together was not associated with a further increase in SIV coverage. We speculate that the highest impact of sending emails or text messages on the SIV uptake would be seen in younger age groups (as compared, for example, with people aged ≥75 years, where SIV coverage is relatively high). Indeed, in Europe and Italy, older age is directly associated with higher GP consultation rates and most so-called frequent GP attenders are seniors [32]. In turn, two-thirds of Italian GPs adopt an opportunistic approach by offering SIV during a patient’s unrelated visit [24].

Finally, our study highlighted a decreasing trend in SIV acceptance: compared with the past year [20], the willingness to receive SIV dropped from 48.6% to 46.9%. A similar decreasing trend (from the 2020/21 to 2022/23 seasons) in different target groups has been reported by the official statistics in both Italy [7] and the US [33]. An initial increase observed in the 2020/21 season is likely driven by a higher effectiveness and reachability of the SIV campaign during the first months of the COVID-19 pandemic when no COVID-19 vaccines were available and there were concerns regarding SARS-CoV-2 and influenza virus co-circulation [20,34]. It has been suggested [35,36] that the subsequent decrease in the 2021/22 and 2022/23 SIV uptake may be linked to safety concerns and mistrust of COVID-19 vaccines, which resulted in a more pronounced hesitancy toward SIV. Pascucci et al. [36] proposed that when COVID-19 vaccines had become available, some individuals expressed concerns over the administration of both SIV and COVID-19 vaccines within a short period and thus prioritized COVID-19 vaccination. Finally, it could also be that the 2021/22 and 2022/23 SIV promotional campaigns were less effective than that conducted during the unprecedented 2020/21 season and therefore SIV coverage rates started to return to the pre-pandemic levels. In summary, there is an urgent need to implement effective strategies able to reverse this negative trend. Our results suggest that personal reminders, preferably sent by GPs via digital channels, may be of aid.

We noted three main study shortcomings that may affect the study results and their interpretation. The first limitation, which may affect the representativeness of the study sample, is the self-reported SIV status that may have induced exposure misclassification bias. On the one hand, a validation study by King et al. [37] has demonstrated a high agreement (97.7% and 93.2% for the current and prior seasons, respectively) between the self-disclosed and registered SIV uptake. On the other hand, it has also been shown [38] that while sensitivity of the self-reported SIV is as high as 100%, its specificity is substantially lower (79%). In other words, some people may overreport their actual SIV uptake owing to recall or social desirability biases. Indeed, SIV coverage observed in our study was higher than that officially reported (20.5%) [7] and this was primarily driven by working-age adults. A similar discrepancy has been reported in another large Italian web-based survey [34]. Another possible explanation may be that the out-of-pocket private purchase of vaccines (i.e., healthy adults for whom no reimbursement is currently provided) could not be registered in the official workflows. Secondly, as in all web-based surveys, our results may be prone to coverage bias due to the digital divide and therefore may not be representative of adults with no Internet access. While we have almost no concerns regarding working-age adults, older adults and especially the oldest old (≥75 years, 7.6% of the whole sample) in our sample of Internet users may systematically differ from non-users. The relationship between Internet use and SIV uptake appears complex. In the US [39], compared with non-users, those who use the Internet but not for health information have 8% (aOR 0.92; 95% CI: 0.88–0.96) decreased odds of being immunized with SIV. No difference (aOR 1.01; 95% CI: 0.97–1.05) between non-users and subjects who used the Internet for informal health information only has been found. Moreover, users who searched the Internet for formal or informal health information were more likely to get vaccinated than non-users (aOR 1.52; 95% CI: 1.45–1.59) [39]. Thirdly, for ethical considerations, we were not able to collect data and perform separate analyses stratified by the presence of single co-morbidities. In particular, this may be relevant to working-age adults, as in this population group, the free-of-charge SIV is offered to subjects with co-morbidities only. Future research should cover this specific population target.

## 5. Conclusions

The results of this representative survey suggest that vaccination reminders may contribute to contrasting the recently observed decline in SIV coverage rates. Reminders sent by a GP, who is the main and most influential source of SIV-related information, and using digital channels like text/instant messaging or emails may have the greatest impact on vaccine uptake.

## Figures and Tables

**Figure 1 vaccines-11-01601-f001:**
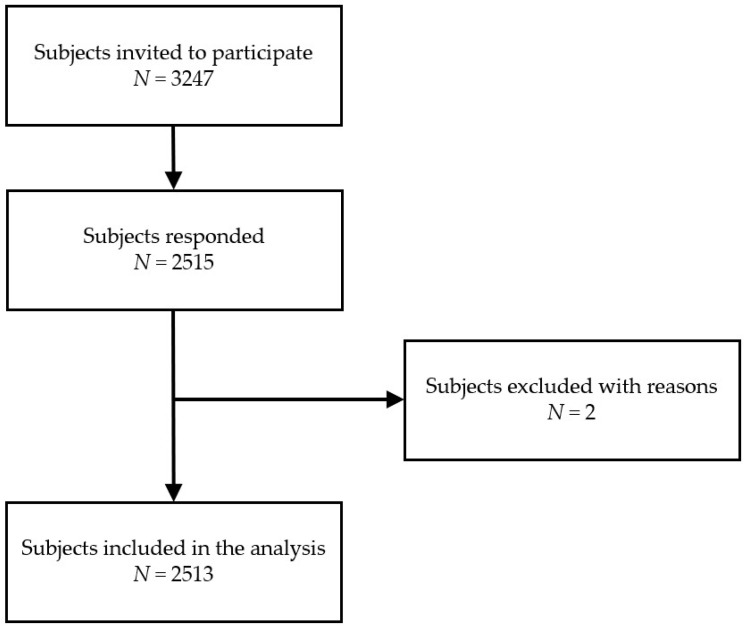
Flowchart of the study participants.

**Figure 2 vaccines-11-01601-f002:**
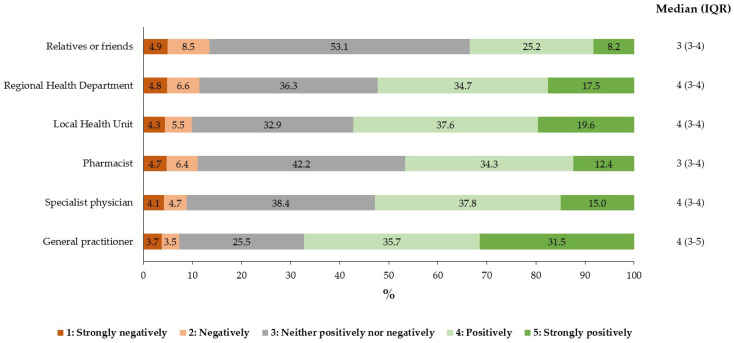
Distribution of the participants’ judgments on the sources of personal reminders to get influenza vaccination (*N* = 2513).

**Table 1 vaccines-11-01601-t001:** Sociodemographic characteristics of the study participants (*N* = 2513).

Characteristic	Level	% (*n*)	95% CI
Sex	Female	52.0 (1307)	50.0–54.0
	Male	48.0 (1206)	46.0–50.0
Age, years	18–24	8.2 (205)	7.1–9.3
	25–34	12.7 (318)	11.4–14.0
	35–44	15.6 (392)	14.2–17.1
	45–54	19.3 (485)	17.8–20.9
	55–64	16.9 (424)	15.4–18.4
	65–74	19.8 (497)	18.2–21.4
	≥75	7.6 (192)	6.6–8.8
Geographic area	North	46.2 (1162)	44.3–48.2
	Center	20.1 (504)	18.5–21.7
	South	33.7 (847)	31.9–35.6
Education level	Low	9.9 (249)	8.8–11.1
	Medium	48.8 (1226)	46.8–50.8
	High	41.3 (1038)	39.4–43.3
Occupation status	Employed	56.4 (1417)	54.4–58.3
	Student	6.8 (172)	5.9–7.9
	Housekeeper	7.8 (197)	6.8–9.0
	Retired	23.1 (580)	21.4–24.8
	Unemployed	4.4 (110)	3.6–5.3
	Other	1.5 (37)	1.0–2.0
Perceived income	Low	2.9 (72)	2.3–3.6
	Lower than average	8.0 (202)	7.0–9.2
	Average	33.2 (834)	31.4–35.1
	Higher than average	38.5 (967)	36.6–40.4
	High	1.7 (43)	1.2–2.3
	No personal income	15.7 (395)	14.3–17.2
2021/22 influenza vaccination	No	53.6 (1348)	51.7–55.6
	Yes	46.4 (1165)	44.4–48.3

**Table 2 vaccines-11-01601-t002:** Association between previous receipt of a reminder to get vaccinated and self-reported influenza vaccination in the 2021/22 season, by source of invitation (*N* = 2513).

Received Invitation to Get Vaccinated From	Level	Vaccinated, % (*n*)	Non-Vaccinated, % (*n*)	OR (95% CI)	aOR (95% CI) ^1^
General practitioner	No	36.9 (430)	81.3 (1096)	Ref	Ref
	Yes	63.1 (735)	18.7 (252)	7.43 (6.20–8.91)	4.43 (3.60–5.48)
Specialist physician	No	73.6 (858)	92.5 (1247)	Ref	Ref
	Yes	26.4 (307)	7.5 (101)	4.42 (3.47–5.62)	2.23 (1.64–3.05)
Pharmacist	No	82.0 (955)	94.1 (1269)	Ref	Ref
	Yes	18.0 (210)	5.9 (79)	3.53 (2.69–4.37)	1.16 (0.80–1.67)
Local health unit	No	80.3 (936)	92.7 (1250)	Ref	Ref
	Yes	19.7 (229)	7.3 (98)	3.12 (2.43–4.01)	1.54 (1.09–2.18)
Regional health department	No	81.2 (946)	93.0 (1254)	Ref	Ref
	Yes	18.8 (219)	7.0 (94)	3.09 (2.39–3.99)	1.11 (0.78–1.59)
Relatives or friends	No	68.2 (794)	85.1 (1147)	Ref	Ref
	Yes	31.8 (371)	14.9 (201)	2.67 (2.20–3.24)	1.26 (0.98–1.62)

^1^ Adjusted for sex, age group, area of residence, education level, perceived income, and other invitation sources; aOR, adjusted odds ratio; CI, confidence interval; OR, odds ratio.

**Table 3 vaccines-11-01601-t003:** Association between the likelihood of receiving the 2022/23 season influenza vaccination and attractiveness of single reminder sources to get vaccinated (*N* = 2513).

Received Invitation to Get Vaccinated From (Reference Category = No)	Model 1 ^1^ aOR (95% CI) ^2^	Model 2 ^3^ aOR (95% CI) ^2^
General practitioner	1.81 (1.44–2.28)	1.68 (1.32–2.15)
Specialist physician	1.00 (0.79–1.27)	1.12 (0.87–1.45)
Pharmacist	1.09 (0.86–1.38)	1.30 (1.02–1.68)
Local health unit	1.51 (1.17–1.95)	1.42 (1.08–1.88)
Regional health department	1.16 (0.92–1.47)	0.98 (0.76–1.28)
Relatives or friends	1.10 (0.89–1.36)	1.03 (0.83–1.27)

^1^ 5-point Likert-based outcome variable of the likelihood of receiving the 2022/23 season dichotomized and coded as (0) for the responses “I don’t know”, “Probably not”, and “Definitely not” and (1) for the responses “Probably yes” and “Yes definitely”; ^2^ adjusted for sex, age group, area of residence, education level, perceived income, past season influenza vaccination, previous receipt of invitations to get vaccination; ^3^ 5-point Likert-based outcome variable of the likelihood of receiving the 2022/23 season dichotomized and coded as (0) for the responses “I don’t know”, “Probably not”, and “Definitely not” and “Probably yes” and (1) for the response “Yes definitely”; aOR: adjusted odds ratio; CI, confidence interval.

**Table 4 vaccines-11-01601-t004:** Preferred communication channels to be invited to get influenza vaccination overall and by age group (*N* = 2513).

Communication Channel	Total (*N* = 2513)		18–59 Years (*N* = 1623)		≥60 Years (*N* = 890)	
	% (*n*)	95% CI	% (*n*)	95% CI	% (*n*)	95% CI
Phone call	8.6 (216)	7.5–9.8	8.3 (135)	7.0–9.8	9.1 (81)	7.3–11.2
Postal letter	17.0 (427)	15.5–18.5	18.0 (292)	16.2–19.9	15.2 (135)	12.9–17.7
Email	27.2 (683)	25.5–29.0	27.7 (449)	25.5–29.9	26.3 (234)	23.4–29.3
Text/instant message	24.6 (618)	22.9–26.3	21.6 (351)	19.6–23.7	30.0 (267)	27.0–33.1
I do not want to receive any invitation	22.6 (569)	21.0–24.3	24.4 (396)	22.3–26.6	19.4 (173)	16.9–22.2

## Data Availability

All relevant data are within the article. Further details may be obtained from the corresponding author upon a reasonable request and prior permission of the Study Funder.

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
