# Peer review of "Increasing Influenza Vaccination Uptake by Sending Reminders: A Representative Cross-Sectional Study on the Preferences of Italian Adults"

_vaccines, 2023, doi:10.3390/vaccines11101601_

Round 1

Reviewer 1 Report

This interesting paper is about the role of reminders in increasing seasonal influence vaccination.

Things to be done.

1.            I think the objectives and hypotheses section should be removed from the material and methods and put inside the introduction.

2.            The authors don't directly explain how the pandemic affected the perceptions around SIV or why there was a subsequent decrease after the initial increase.

3.            Indicate any specific R packages or libraries utilized beyond the R base stats package.

4.            The word "noting" in "do noting" strategies should be " do nothing." (line 278)

5.            There's a comparison between the SIV acceptance rate between two years. Write a hypothesis or potential reasons for this decrease.

Author Response

Comment: This interesting paper is about the role of reminders in increasing seasonal influence vaccination. Things to be done.

Reply: Thank you for your interest in our paper. All your comments have been now addressed.

Comment: 1. I think the objectives and hypotheses section should be removed from the material and methods and put inside the introduction.

Reply: As suggested, we have now moved these sections to the “Introduction”.

Comment: 2. The authors don't directly explain how the pandemic affected the perceptions around SIV or why there was a subsequent decrease after the initial increase.

Reply: Thank you for this comment. We have now discussed on how the pandemic impacted SIV perceptions and coverage rates and described some reasons of the observed decrease after the initial increase.

Comment: 3. Indicate any specific R packages or libraries utilized beyond the R base stats package.

Reply: The packages used have been added.

Comment: 4. The word "noting" in "do noting" strategies should be " do nothing." (line 278)

Reply: This typo has been corrected.

Comment: 5. There's a comparison between the SIV acceptance rate between two years. Write a hypothesis or potential reasons for this decrease.

Reply: As suggested, we have identified and discussed three main reasons for the observed decrease in SIV acceptance.

Reviewer 2 Report

General comment

The paper “Increasing Influenza Vaccination Uptake by Sending 2 Reminders: A Representative Cross-Sectional Study on the 3 Preferences of Italian Adultsis an interesting and well-written article on the experience and attitudes of Italian adults towards the active invitation to receive the flu vaccine, promoted by different sources and transmitted by different communication channels, and to evaluate the projected benefits of this strategy; but the paper need editorial review.

Major comments

The authors must explain and describe the intention to get vaccinated against influenza based on whether or not they have indications (chronic diseases, healthcare personnel, etc.) to get vaccinated. Participants are more likely to have received vaccination souvenirs the previous season if they are from a risk group!, Please, comment on.

The authors must explain and describe the intention to get vaccinated against influenza based on whether or not they have indications (chronic diseases, healthcare personnel, etc.) to get vaccinated. Additionally, they should better explain in Methods and Results the total number of survey candidates (10,000?), the final response rate, and compare those who responded with those who did not respond (attach a flowchart). Comment on limitations to the problems of representativeness of the exhibition.

Specific comments

1)      Introduction: Explain whether doctors' vaccination reminders are made to the entire population or only to risk groups.

2)      In Methods explain total number of survey candidates (10,000?), the final response rate.

3)      Explain intention to get vaccinated and preferred communication channel by age groups and flu risk groups

4)      Evaluate and comment in the first paragraph of the Discussion on the two hypotheses studied in a concise manner.

5)      Comment in more detail on response rates and limitations of representativeness.

Author Response

Comment: The paper “Increasing Influenza Vaccination Uptake by Sending 2 Reminders: A Representative Cross-Sectional Study on the 3 Preferences of Italian Adults” is an interesting and well-written article on the experience and attitudes of Italian adults towards the active invitation to receive the flu vaccine, promoted by different sources and transmitted by different communication channels, and to evaluate the projected benefits of this strategy; but the paper need editorial review.

Reply: Thank you for your interest in our paper. All your comments have been now addressed.

Comment: The authors must explain and describe the intention to get vaccinated against influenza based on whether or not they have indications (chronic diseases, healthcare personnel, etc.) to get vaccinated. Participants are more likely to have received vaccination souvenirs the previous season if they are from a risk group!, Please, comment on.

Reply: Thank you for this comment. For ethical reasons, we were not able to collect data on particular chronic conditions, for which annual vaccination is recommended. To circumvent this issue, all models have been adjusted for age and previous season vaccination. Indeed, these two variables are the main determinants of the current season influenza vaccination and may sufficiently adjust for vaccine eligibility. However, on considering that in Italy influenza vaccination is currently recommended for all older adults aged ≥60 years, we how now performed a subgroup analysis by age group (see also your comments below). Moreover, we have now clearly stated among the study limitations that we were not able to perform separate analyses stratified by the presence of co-morbidities and this issue should be addressed in future research.

Comment: The authors must explain and describe the intention to get vaccinated against influenza based on whether or not they have indications (chronic diseases, healthcare personnel, etc.) to get vaccinated.

Reply: As suggested, we have now reported intentions to get vaccinated in adults aged ≥60 years (who are all eligible for a free-of-charge vaccination) and those aged 18–59 years. Unfortunately, for ethical reasons, we were not able to collect data on particular chronic conditions and this has been clearly indicated among the study limitations (see also our reply above).

Comment: Additionally, they should better explain in Methods and Results the total number of survey candidates (10,000?), the final response rate, and compare those who responded with those who did not respond (attach a flowchart). Comment on limitations to the problems of representativeness of the exhibition.

Reply: The total number of survey candidates for this cross-sectional round of a longitudinal panel was 3247, while the response rate was 77.46%. This has been clearly stated. We agree that the description of sampling strategy may be a bit confusing. We have therefore amended this part of Methods. The sample was judged representative of the Italian adult population from the point of view of principal socio-demographic characteristics. The sample may have some issues on representativeness from the point of view of the declared influenza vaccination uptake as well as we cannot exclude the presence of the digital-divide bias. We have further enhanced this in describing the study limitations. As suggested, a flowchart of the study participants has been added and a comparison between responders and non-responders has been provided.

Comment: 1) Introduction: Explain whether doctors' vaccination reminders are made to the entire population or only to risk groups.

Reply: In Italy, influenza vaccination is recommended for anyone aged 6+ months. However, it is currently reimbursed only for older adults aged ≥60 years, pregnant women, subjects with underlying health conditions, children aged 6 months to 6 years, some categories of workers. This has been now clearly stated. We have also specified which population groups were targeted by reminders in the available Dutch UK studies.

Comment: 2) In Methods explain total number of survey candidates (10,000?), the final response rate.

Reply: This has been done ((see also our reply above).

Comment: 3) Explain intention to get vaccinated and preferred communication channel by age groups and flu risk groups

Reply: As required, we have now compared (in both main text and Table 4) intentions to get vaccinated and preferred communication channels between younger and older adults. As we described above, we were nota able to collect data on co-morbidities.

Comment: 4) Evaluate and comment in the first paragraph of the Discussion on the two hypotheses studied in a concise manner.

Reply: Thank you for this suggestion. We have now briefly described our results with reference to the pre-specified hypotheses.

Comment: 5) Comment in more detail on response rates and limitations of representativeness.

Reply: As we also mentioned earlier, while the overall sample was judged representative of the Italian adult population from the point of view of principal socio-demographic characteristics, representativeness may be affected by a different influenza vaccination uptake (found in the study vs officially reported). We also cannot exclude the presence of the digital-divide bias. These issues have further enhanced among the study limitations. 

Round 2

Reviewer 2 Report

Author have done the modifications and the paper may be accepted